# Chronic Stress Segregates Mice into Distinct Behavioral Phenotypes Based on Glucocorticoid Sensitivity

**DOI:** 10.3390/ijms262311436

**Published:** 2025-11-26

**Authors:** Polina Ritter, Rasha Salman, Yuliya Ryabushkina, Natalya Bondar

**Affiliations:** 1Institute of Cytology and Genetics (ICG), Siberian Branch of Russian Academy of Sciences (SB RAS), Prospekt Akad. Lavrentyeva 10, 630090 Novosibirsk, Russia; p.kisaretova@gmail.com (P.R.); salmanrasha030@gmail.com (R.S.); ryabushkina@bionet.nsc.ru (Y.R.); 2Department of Natural Sciences, Novosibirsk State University, Pirogova Street 2, 630090 Novosibirsk, Russia

**Keywords:** chronic social defeat stress, glucocorticoids, HPA axis

## Abstract

Chronic stress alters hypothalamic–pituitary–adrenal (HPA) axis function, affecting corticosterone regulation and adaptive responses. Understanding individual variability in stress adaptation requires identifying distinct HPA axis response patterns. Here, we assessed HPA axis sensitivity in male C57BL6 mice exposed to 30 days of chronic social defeat stress (CSDS). Negative feedback integrity was evaluated using the dexamethasone suppression test (DST), with corticosterone measured after saline or low-dose dexamethasone administration at days 10 and 30. Behavioral testing (open field, elevated plus maze, social interaction test, partition, social defeat, forced swimming test, sucrose preference test) and qPCR analysis of HPA-axis-related genes in the hypothalamus (*Crh*, *Crhr1*, *Crhbp*, *Fkbp5*, *Nr3c1*), pituitary (*Pomc*, *Crhr1*, *Nr3c1*, *Nr3c2*), and adrenal glands (*Cyp11a1*, *Cyp11b1*, *Hsd11b1*, *Mc2r*, *Star*, *Fkbp5*, *Nr3c1*) were performed. K-means cluster analysis identified three distinct response profiles differing in baseline and dexamethasone-suppressed corticosterone levels. Clusters also exhibited differences in behavioral phenotypes and HPA axis gene expression. Cluster 1 showed low basal corticosterone and an abnormal dexamethasone suppression response, without significant *Crh* or *Crhbp* dysregulation in the hypothalamus. Cluster 2 exhibited elevated basal corticosterone, a blunted dexamethasone response, anhedonia, and reduced immobility in the forced swim test; increased *Crh* and reduced *Fkbp5* suggested enhanced glucocorticoid receptor sensitivity and sustained hypercortisolemia. Cluster 3, characterized by normal basal corticosterone and normal dexamethasone response, displayed upregulation of *Crh* and *Crhbp*, consistent with balanced and potentially adaptive HPA axis regulation under chronic stress. These results demonstrate that corticosterone response heterogeneity reflects distinct adaptive trajectories under chronic stress. Identifying behavioral and molecular markers of these strategies may advance understanding of stress vulnerability and resilience mechanisms, with implications for stress-related disorders.

## 1. Introduction

Chronic stress is known to dysregulate the HPA axis, leading to altered secretion of glucocorticoids such as corticosterone in rodents and cortisol in humans [1].

Chronic social defeat stress (CSDS) is commonly employed to model stress-related disorders such as depression in animals. As a stress-related illness, major depressive disorder (MDD) is also associated with an abnormal HPA axis response to stress. Both people with depression and animals after CSDS typically show alterations in the sensitivity and amplitude of the HPA axis response, altered diurnal patterns of hormone secretion, and resistance of the immune system to glucocorticoids [2,3,4,5]. Elevated basal glucocorticoid levels are a key indicator of HPA axis hyperactivity, a condition frequently seen in both stressed animals and depressed individuals [6,7,8]. Elevated corticosterone levels in the CSDS model are shown starting from 10 days of stress [9], after 30 days [7], and an even extended protocol of CSDS for 7 weeks showed elevated morning corticosterone levels in male mice [10].

To evaluate the feedback mechanism of the HPA axis, the dexamethasone suppression test (DST) is often used. Low-dose treatment with dexamethasone, a synthetic glucocorticoid, typically suppresses intrinsic corticosterone/cortisol secretion. In chronically stressed animals, this suppression may be incomplete or delayed. Short-term social stress (7 days) in male rats had no effect on dexamethasone suppression sensitivity [11]. However, after 21 days of social stress, weaker suppression by dexamethasone was observed in rats [12] and CD1 mice [13]. This impaired feedback mechanism is a significant marker of HPA axis dysregulation, which parallels findings in humans with depression, where cortisol suppression is often incomplete following dexamethasone administration [14,15].

Interestingly, there is substantial individual variability in how both humans and animals respond to the DST [16], indicating that while HPA axis dysregulation is a consistent finding in chronic stress and depression, the exact nature of the dysregulation can differ. Some individuals may show a complete resistance to glucocorticoid feedback, while others exhibit only partial suppression, suggesting that the degree of HPA axis impairment may vary depending on factors such as stress duration, severity, and individual vulnerability.

Similarly, rodent studies show that different stress paradigms impair the HPA axis in distinct ways. For example, after 14 days of foot shock, rats exhibited impaired glucocorticoid feedback and higher corticosterone secretion in response to acute stress [17], but not after 7 days of foot shock. Conversely, 14 days of forced swim did not affect the amplitude of the corticosterone response to acute stress or feedback inhibition. Chronic dexamethasone treatment also showed variable effects: 14 days of dexamethasone administration impaired negative feedback without exaggerating the response to an acute stressor in rats [17]. In one study, 21 days of dexamethasone treatment led to low basal corticosterone in C57BL/6J mice [18]. These findings indicate that both the severity and duration of stress play a critical role in modulating the HPA axis.

Moreover, individual variability in how mice respond to chronic stress is reported in studies. C57BL/6 mice are often categorized as either stress-sensitive or stress-resistant, with reported differences in behavior and even neurobiological adaptations [19,20,21]. For example, one study found that resilient C57BL/6J mice displayed increased glucocorticoid receptor (*Nr3c1*) expression and nuclear translocation in the hippocampus [22], suggesting this mechanism is crucial for stress resilience. Another study found that increased plasma corticosterone levels after stress predicted susceptibility to depression-like behaviors [20]. Genomic studies have shown that epigenetic modifications play an important role in distinguishing resilience from susceptibility to CSDS. In a study by Kuehner et al. [23], alterations in 5mC and 5hmC levels in the brain were found to differentiate resilient mice, which displayed neuroprotective molecular signatures, from susceptible mice, which exhibited markers of neurodegeneration. Similarly, a study by O’Toole et al. (2019) [24] demonstrated differential DNA methylation patterns after CSDS, with CG hypermethylation being more prominent in susceptible mice. Genes such as *Esr1* and *Cacna1c*, linked to psychiatric conditions, were among those with the most differentially methylated sites [24].

Beyond stress vulnerability, intrinsic variability, even in isogenic populations, is well-documented [25]. This variation underscores the importance of understanding the heterogeneity in stress responses, particularly in relation to the HPA axis, which may help elucidate why some individuals or animals are more resilient to stress, while others are more susceptible to its negative effects.

Research consistently demonstrates that chronic stress induces HPA axis dysregulation at multiple levels. For instance, stress can enhance *Crh* expression in the hypothalamus [26,27], which drives the HPA axis to secrete more glucocorticoids and reduce glucocorticoid receptor expression in key brain regions involved in feedback regulation, such as the hippocampus and prefrontal cortex, contributing to disrupted feedback regulation. While the brain is primarily affected, chronic stress also impacts peripheral tissues, especially those involved in glucocorticoids synthesis. Changes in adrenal gland size and altered glucocorticoid receptor activity in peripheral tissues contribute to the systemic hormonal dysregulation observed under chronic stress. However, while stress response heterogeneity, such as the distinction between susceptible and resilient mice, has been observed, only a few studies explore the heterogeneity of HPA axis response. As a result, the impact of individual differences on HPA axis dysregulation remains unclear, yet understanding the variability in glucocorticoid sensitivity is important, as this variability may contribute to differences in antidepressant response and help guide personalized treatment choices. This is especially important since depression is a heterogeneous disorder with several phenotypes described in the literature [28], yet all available treatments have similar mechanisms of action and in many cases fail to succeed.

To address these gaps, in this study, we aim to evaluate the variability in the sensitivity of the HPA axis following chronic social defeat stress. We used an extended 30-day social defeat protocol, incorporating dexamethasone suppression tests on both days 10 and 30 to assess transitional and sustained HPA axis states. We increased the sample size to enhance statistical power, allowing us to detect potential variability in corticosterone levels and individual differences in HPA axis response. Moreover, to determine whether differences in HPA axis sensitivity correlate with behavioral traits, we performed a series of behavioral tests (open field, elevated plus maze, social interaction test, partition, social defeat, forced swimming test, sucrose preference test). To uncover the molecular mechanisms linked to different levels of HPA axis sensitivity, we analyzed the expression of genes related to the HPA axis in the hypothalamus, pituitary, and adrenal glands using qPCR.

## 2. Results

To evaluate stress-related behavioral and endocrine alterations, we conducted two independent experiments using CSDS in male C57BL/6 mice.

Experiment 1 (Figure 1A, left) included stressed animals and two control groups (behaviorally tested and intact). Mice underwent CSDS for 31 days, with DST performed after 10 and 30 days. Behavioral assessments (elevated plus maze, open field, social interaction, and partition tests) were conducted during the final week of stress.

Experiment 2 followed a similar design (Figure 1A, right), with DST performed around day 30 and additional behavioral tests, including sucrose preference and forced swim tests. After the last CSDS session, animals were sacrificed for collection of the hypothalamus, pituitary gland, and adrenal glands for qPCR analysis.

### 2.1. Effects of Behavioral Testing on Basal Corticosterone Levels in Control

We used a three-way ANOVA to analyze the effects of experiment, behavioral testing, and dexamethasone administration on corticosterone levels in control animals (Figure 1B). A significant reduction in hormone levels was observed following dexamethasone administration [F(1, 59) = 23.13, *p* < 0.001] in both experiments.

Importantly, there was a significant interaction between the factors of experiment and behavioral testing [F(1, 59) = 12.60, *p* < 0.001], indicating that the impact of behavioral procedures on corticosterone levels depended on the nature and intensity of the tests. Specifically, in Experiment 1, animals exposed to a battery of behavioral tests (elevated plus maze, open field, social interaction, and partition test) had significantly higher corticosterone levels compared to intact animals (*p* < 0.01). In contrast, in Experiment 2, where only the sucrose preference test was used, no significant differences were detected between intact and behaviorally tested controls.

Dexamethasone reduced corticosterone levels regardless of behavioral testing experience, with no significant interaction between these factors [F(1, 59) = 1.49, *p* = 0.23].

Thus, we have demonstrated that behavioral testing affects the basal corticosterone level in control animals. Based on this, to further characterize the results obtained in Experiment 1, we used an intact control group, whereas in Experiment 2, the control groups were combined.

When comparing the basal corticosterone levels of animals in the CSDS groups between Experiment 1 and Experiment 2, no significant differences were found (Figure 1C), despite the variation in the number of behavioral tests conducted between the groups. This suggests that the stress induced by behavioral testing is likely negligible compared to the daily social defeat stress in stressed animals. Therefore, we can conclude that, for chronically stressed groups, participation in behavioral tests does not influence resting corticosterone levels.

### 2.2. HPA Axis State Evaluation After 10 and 30 Days of CSDS with DST

Under prolonged stress, continuous HPA axis activation can lead to elevated basal (resting) glucocorticoid levels, while further HPA activation may result in compensatory suppression of the axis and impaired negative feedback. In this experiment, using the dexamethasone test, we assessed the ability of the glucocorticoid system in mice with CSDS experience to effectively suppress HPA activity in response to stress after administration of the synthetic glucocorticoid dexamethasone.

In Experiment 1, we measured corticosterone levels and their response to dexamethasone injection after 10 and 30 days of CSDS. Basal corticosterone in CSDS groups did not differ from controls and was unaffected by stress duration. We previously reported elevated morning corticosterone after 30 days of CSDS [7], but the difference between groups likely diminishes during the daytime hormone peak. Dexamethasone administration reduced corticosterone in all groups, but suppression was more pronounced in the 10-day stress group (Figure 1C). Two-way ANOVA revealed only a treatment effect [F(1, 109) = 46.1, *p* < 0.001]. In Experiment 2, we observed the same pattern (control vs. CSDS 30 days)—corticosterone reduction 6 h post-dexamethasone regardless of group, with only a significant treatment effect [F(1, 78) = 21.1, *p* < 0.001] (Figure 1C).

### 2.3. Clustering by Corticosterone Response in DST

In both experiments, the group of animals with 30-day CSDS experience exhibited high variability in corticosterone levels. Specifically, the CSDS group showed a standard deviation (SD) of 142 and 124 for Experiment 1 and 2, respectively, whereas the control group had an SD of 107 and 119. The heatmap reveals that by day 10 of stress, most animals showed reduced corticosterone levels in the dexamethasone suppression test. However, by day 30, both experiments included animals with either no change or even elevated corticosterone levels after dexamethasone treatment (Figure 2).

The large sample size in both experiments (23 mice in Experiment 1 and 25 mice in Experiment 2, with sufficient serum collected for corticosterone measurement at both stages of the dexamethasone test) allowed us to perform k-means clustering to identify distinct response patterns to dexamethasone administration and define functional subgroups.

CSDS mice were clustered based on their dexamethasone test results after 30 days of stress (basal corticosterone levels and levels 6 h post-dexamethasone injection). Clustering was performed on Z-scaled values (Figure 3).

In both Experiment 1 and 2, we identified three distinct subgroups (Figure 3A), with nearly identical variation in corticosterone dynamics (Figure 3B,C). (1) Abnormal responders (exp1: 26%, n = 6; exp2: 20%, n = 5): animals showing an atypical increase in corticosterone after dexamethasone, with basal levels consistently below controls. (2) High basal corticosterone (exp1: 30%, n = 7; exp2: 29%, n = 7): mice with elevated basal corticosterone (relative to controls) and only a mild (2-fold) reduction after dexamethasone, regardless of stress duration. (3) Strong suppressors (exp1: 44%, n = 10; exp2: 50%, n = 12): mice with low basal corticosterone and a pronounced (4-fold) dexamethasone-induced suppression, reaching levels comparable to intact controls.

Notably, the proportional distribution of animals across clusters was highly consistent between experiments, despite methodological differences (e.g., blood collection protocols, timing of measurements, and ELISA kits from different manufacturers). This reproducibility strongly suggests that the observed clusters reflect genuine, stress-induced alterations in dexamethasone sensitivity.

### 2.4. Behavior Analysis

To assess stress-induced behavioral changes, we performed a series of standardized tests across two experimental series. In Experiment 1, on days 25–28, animals underwent the elevated plus maze, open field, social interaction and partition tests, with additional analysis of a social defeat session on day 24. Experiment 2 included the sucrose preference test (days 28–30) and the forced swim test (day 34).

Chronic stress affected anxiety-related behavior parameters in the elevated plus maze test (Figure 4A), as shown by one-way ANOVA (factor “group”) with post hoc Tukey’s HSD. Animals from the CSDS group spent significantly less time in the center (*p* < 0.001) and in the open arms (*p* < 0.001), and significantly more time in the closed arms (*p* < 0.001) compared to controls, indicating increased anxiety-like behavior.

In the open field test (Figure 4B) (one-way ANOVA, factor “group”), animals from the CSDS group explored a smaller area of the open field (*p* < 0.001) and spent less time on exploratory behavior (rearing: *p* < 0.001). Additionally, chronically stressed animals spent less time in the center of the field (*p* < 0.001) compared to the control group, confirming the increased anxiety level detected in the elevated plus maze test.

In the partition test (Figure 4C and Appendix A), which assesses social behavior in the home cage under familiar conditions, a two-way ANOVA (factors “group” and “partner type”) revealed significant main effects of both “group” [F(1, 95) = 20.39, *p* < 0.001] and “partner type” [F(1, 95) = 46.72, *p* < 0.001] on time spent near the partition, but no interaction between these factors. Mice exposed to CSDS spent less time near the partition next to a social partner compared to the control, regardless of whether the partner was familiar or unfamiliar (familiar: *p* < 0.01, unfamiliar: *p* < 0.01). Both the control and CSDS groups showed greater interest in the unfamiliar social partner than in the familiar one, spending more time near the partition when the unfamiliar partner was presented (control: *p* < 0.01, CSDS: *p* < 0.001). The latency to approach the partition did not differ significantly between control and CSDS groups. Next, a two-way ANOVA (factors “group” and “partner type”) revealed an effect of the “group” factor on grooming [F(1, 95) = 11.9, *p* < 0.001] and rearing durations [F(1, 95) = 40.1, *p* < 0.001]. Animals from the CSDS group spent more time grooming (*p* < 0.05 when a familiar partner was present, *p* < 0.001 with an unfamiliar) and significantly less time engaged in exploratory activity—rearing (*p* < 0.05 familiar partner, *p* < 0.001 unfamiliar)—compared to control. The “partner type” factor was also significant [F(1, 95) = 91.6, *p* < 0.001], with animals spending less time on rearing activity when an unfamiliar partner was introduced (*p* < 0.05 familiar partner, *p* < 0.001 unfamiliar).

In the social interaction test (Figure 4D and Appendix A), where social behavior was assessed in an unfamiliar neutral cage, a two-way ANOVA (factors “group” and “trial”) revealed effects of the “group” factor [F(1, 90) = 10.9, *p* < 0.01], the “trial” factor [F(1, 90) = 24.4, *p* < 0.001], and their interaction [F(1, 90) = 4.7, *p* < 0.05] on time spent in the target zone, as well as significant effects of the “group” factor [F(1, 87) = 54.4, *p* < 0.001] and the “trial” factor [F(1, 87) = 17.1, *p* < 0.001] on rearing duration. Animals with chronic stress experience approached the box with a social partner significantly less often (second trial of the test) than control animals (*p* < 0.01), although their interest in the social partner was still significantly higher than in the empty box during the first trial (*p* < 0.05). The latency to approach the target zone did not differ between trials or groups. In this test, we also observed a decrease in exploratory activity (rearing) in the control group when a social partner was presented (trial 2, *p* < 0.01), which was not observed in the CSDS group, where exploratory activity was reduced compared to the control in both trials (trial 1: *p* < 0.001, trial 2: *p* < 0.01).

The forced swim test (Figure 4E) showed increased depressive-like behavior in the CSDS group, with significantly reduced active time (*p* < 0.05) and increased immobility time (*p* < 0.01) compared to the control.

The sucrose preference test (Figure 4F and Appendix A) did not reveal significant changes between control and CSDS groups.

Thus, chronic stress exposure led to anxiety- and depression-like behaviors, as well as reduced social activity in male C57BL/6 mice, which is consistent with previous studies [7,13,29,30] and confirms the validity of the chronic stress model used in the experiments.

### 2.5. Behavioral Characteristics of Animals with Different HPA Axis Response in DST

We aimed to identify behavioral markers that could reflect cluster segregation in CSDS mice (Figure 4). Analysis of behavior in animals clustered based on their dexamethasone response revealed notable patterns.

Mice from the first cluster (~20% of animals), exhibiting an abnormal increase in corticosterone levels in the DST, displayed the lowest social activity in the partition test (toward both familiar and unfamiliar partners). However, in the forced swim test, their behavior was nearly indistinguishable from controls, whereas other subgroups (clusters 2 and 3) showed increased immobility. Notably, during confrontation with an aggressor, mice from this cluster exhibited more freezing behavior than others, with freezing persisting even after the aggressor retreated and contact ceased (Figure 4G).

Mice from the second cluster (~30% of animals), characterized by elevated basal corticosterone levels and a weak dexamethasone suppression, developed anhedonia (reduced sucrose preference, *p* < 0.001 vs. control), decreased immobility time (*p* < 0.01) in the forced swim test, and control-like social activity levels in the home cage (partition test).

The third cluster (~50% of animals) did not exhibit any distinct behavioral features, displaying a typical stress-induced phenotype. Their behavior largely mirrored that of the overall CSDS group, showing reduced time spent in the open arms of the elevated plus maze and in the center of the open field, along with a smaller explored area. In the forced swim test, these mice exhibited increased immobility and a shorter latency to immobility. During the social defeat test, they demonstrated more frequent freezing near the aggressor and spent more time away from it, and in the partition test, they spent less time near the partition.

Thus, mice with an abnormal dexamethasone response (cluster 1) demonstrated a more pronounced passive defense strategy (freezing), accompanied by reduced social behavior even in the neutral home cage environment. At the same time, these animals did not exhibit pronounced depressive-like behavior in classical depression tests (sucrose preference and forced swim test). Mice from the second cluster, with high basal corticosterone levels, displayed characteristic depression-like behaviors while maintaining a control-like response in the dexamethasone suppression test. In contrast, mice with a control-like dexamethasone response (cluster 3) exhibited a typical CSDS-like phenotype, characterized by anxiety and depression-like behaviors such as reduced exploration, increased immobility and enhanced avoidance of the aggressor.

### 2.6. qPCR Results

Additionally, in Experiment 2, we performed qPCR gene expression analysis on key HPA axis genes in the hypothalamus (*Crh*, *Crhr1*, *Crhbp*, *Fkbp5*, *Nr3c1*), pituitary gland (*Pomc*, *Crhr1*, *Nr3c1*, *Nr3c2*), and adrenal glands (*Cyp11a1*, *Cyp11b1*, *Hsd11b1*, *Mc2r*, *Star*, *Fkbp5*, *Nr3c1*), and additionally analyzed *Gh* (growth hormone) and *Prl* (prolactin) in the pituitary gland (Figure 5).

Chronic 30-day stress had a significant effect on expression of the central HPA axis regulator *Crh* [F(3, 31) = 4.425, *p* < 0.05] in the hypothalamus and its binding protein *Crhbp* [F(3, 31) = 4.754, *p* < 0.01]; however, this effect varied between clusters. Compared to control, *Crh* was upregulated in clusters 2 (*p* < 0.05) and 3 (*p* < 0.05), while *Crhbp* was significantly upregulated only in cluster 3 (*p* < 0.01). Interestingly, cluster 1 did not significantly differ from the control.

In the pituitary gland, we observed no significant effects of stress on gene expression.

We showed that changes in expression of genes coding for corticosterone synthesis enzymes in the adrenal glands are present in all stressed animals regardless of cluster. Such changes are typical for animals after chronic stress exposure [31] (upregulation of *Cyp11a1* [F(3, 32) = 12.14, *p* < 0.001], *Cyp11b1* [F(3, 32) = 10.46, *p* < 0.001] and *Star* [F(3, 32) = 7.613, *p* < 0.001]; downregulation of *Mc2r* [F(3, 32) = 18.2, *p* < 0.001]).

Downregulation of *Fkbp5* [F(3, 32) = 3.887, *p* < 0.05] in the adrenal glands was present only in cluster 2 (*p* < 0.05) compared to the control. For remaining genes, we did not detect significant changes in expression.

## 3. Discussion

### 3.1. Dynamics of HPA Axis Responsiveness During Chronic Social Stress

In this study, we examined how chronic social stress affects the response to dexamethasone administration, assessing HPA axis activity in C57BL/6 mice and its ability to respond to glucocorticoid stimulation. These findings provide insights into how chronic stress and glucocorticoids impact the molecular pathways that regulate stress adaptation and neurobiological functions.

Chronic social stress affects corticosterone levels depending on the duration of stress exposure. Initially, it increases due to repeated acute stress responses, but with continued exposure, corticosterone levels normalize or even decrease as a result of HPA axis adaptation. The results of the dexamethasone suppression test reflect these changes: in the early phase of chronic stress (10 days), the negative feedback system remains resistant to glucocorticoid elevation, whereas with prolonged stress, the regulation either becomes impaired or remains adaptive depending on individual variability in stress responsiveness. This dynamic is critical for understanding the physiological and psychological consequences of chronic stress and has implications for the treatment of stress-related disorders [32,33].

Our results show that mice subjected to 10 days of chronic social stress, similar to control animals, have significantly reduced corticosterone levels in response to dexamethasone administration (by 76% for 10 days and 73% for 30 days). However, 30 days of stress leads to a much weaker response—a reduction of only 20%. This indicates that despite prolonged stress, the glucocorticoid feedback mechanism required to limit the stress response remains functional in the early stages of stress exposure. Nevertheless, Buwalda et al. [34] demonstrated that even brief episodes of social stress (2 days) can alter HPA axis regulation, as dexamethasone reduced corticosterone levels in stressed rats, yet ACTH remained elevated, indicating that a reduction in pituitary sensitivity to glucocorticoids in stressed animals begins early and possibly intensifies with chronic exposure.

### 3.2. Heterogeneity of Stress Responses and Their Possible Correspondence to MDD Subtypes

The variation we observed in corticosterone levels (both baseline and dexamethasone-suppressed) by day 30 in the CSDS group may reflect progressive adaptation or dysregulation of the HPA axis under prolonged stress exposure. This variability supports clinical findings in depressed patients, suggesting potential HPA axis dysregulation under chronic stress. Individuals with severe depression or chronic stress often show altered cortisol regulation, which may manifest as either an exaggerated or blunted cortisol response to stress [35,36].

Different subtypes of MDD have been proposed to account for variability in HPA axis activity, particularly the division into melancholic and atypical subtypes [37]. Melancholic depression, characterized by low mood, anhedonia, loss of appetite, weight loss, and insomnia, is associated with elevated basal cortisol levels and a stronger cortisol response in the dexamethasone suppression test compared to atypical depression. Some patients also exhibit impaired HPA axis inhibition (i.e., lack of cortisol suppression) via negative feedback [37]. In contrast, atypical MDD is marked by heightened emotional reactivity to external stimuli and at least two of the following: overeating, hypersomnia, leaden paralysis, and dysphoria. In this subtype, basal cortisol levels are either normal or reduced, and the dexamethasone suppression test typically shows either no significant differences from controls or a more pronounced suppression [37].

Before extending these observations to our findings, it is important to stress that this comparison is conceptual rather than translational. Mouse behavioral assays capture only a limited set of depressive-like features, and our neuroendocrine measures focus on corticosterone dynamics and selected HPA axis genes. Therefore, any analogy with human melancholic or atypical depression should be viewed as a way to contextualize variability in stress responses, not as evidence of subtype-specific pathology in mice.

In this context, the subgroups of stressed animals that we identified based on their dexamethasone response may show parallels with the depression subtypes described above. For example, a group of mice showing an abnormal (increased) corticosterone response to dexamethasone and reduced social activity may resemble atypical depression, where paradoxical responses to stress and medication are common [38,39] and where social withdrawal is prominent [40], despite milder classical depressive symptoms [41].

A second cluster of stressed animals, characterized by high basal corticosterone and anhedonia, may resemble melancholic depression, which is marked by elevated physiological stress markers (e.g., high cortisol) and reduced capacity for pleasure. Elevated corticosterone, alongside increased immobility in the forced swim test, may reflect neurobiological adaptations to chronic stress, akin to the psychomotor retardation seen in melancholic individuals despite high stress hormone levels [36].

A third cluster, which shows typical stress-induced behavior and a dexamethasone response indistinguishable from controls, may reflect features of mild or moderate depressive episodes, where stress reactivity is evident but without pronounced neuroendocrine dysfunction.

Studies using the same stress model have shown that mice exposed to chronic social stress and exhibiting passive behavior during confrontations displayed elevated corticosterone levels by day 21, which then dropped below baseline after the cessation of daily confrontations [42]. In contrast, animals with active coping behavior during confrontations had basal corticosterone levels comparable to controls. Active coping was also associated with more stable sympatho-adrenomedullary axis activity, as indicated by higher catecholamine levels compared to passively coping animals.

### 3.3. Transcriptional Signatures Associated with Differential HPA Axis Responses

Our findings of variable behavioral and corticosterone responses point to the underlying heterogeneity that frequently manifests during prolonged stress exposure. Even organisms as simple as yeast, exposed to uniform conditions, display a wide range of phenotypic responses to environmental changes, as shown by single-cell RNA-seq data [25]. This heterogeneity likely arises from the internal state of cells [25], which can be reflected in transcript and protein levels as well as epigenetic modifications and is affected by random molecular events and environmental factors. Although this phenotypic variability is often temporary, it can still direct long-term outcomes.

Genes involved in vital processes, such as housekeeping genes, show less variation in expression levels, while stress-responsive genes display higher variability. This may be because fluctuations in stress-related proteins help cells adapt to changing environments [43]. Stress has been shown to significantly impact heterogeneity, increasing noise levels in gene expression, protein production, and physiological responses, as seen in both yeast [44] and mammalian studies [45].

The diverse patterns in behavioral responses and corticosterone levels we observed underscore the individual variability that chronic stress tends to amplify. qPCR provides an additional layer of information, revealing the molecular underpinnings of variability in HPA axis regulation in animals exposed to chronic social stress.

In cluster 1, animals exhibit abnormal corticosterone elevation in the dexamethasone suppression test, indicating impaired negative feedback regulation of the HPA axis. Consistent upregulation of adrenal corticosterone-synthesis genes (*Cyp11a1*, *Cyp11b1*, *Star*) reflects an increased capacity for corticosterone production that is only activated under stress or challenge conditions. The absence of significant *Crh* or *Crhbp* dysregulation in the hypothalamus suggests that central HPA axis regulators are not the primary drivers of this phenotype, and that adrenal dysregulation plays a critical role.

Cluster 2 displayed elevated basal corticosterone, anhedonia, and reduced immobility in the forced swim test, suggesting a hyperactive HPA axis state with altered coping strategies. Anhedonia, as evidenced by reduced sucrose preference, may result from disruptions in reward system pathways sensitive to chronic corticosterone exposure and glucocorticoid receptor activity. Molecularly, elevated *Crh* promotes HPA axis hyperactivation, while reduced *Fkbp5* may enhance glucocorticoid receptor sensitivity, leading to prolonged transcriptional effects of corticosterone and likely contributing to the observed hypercortisolemia. In humans and animal models, there is evidence of *Fkbp5* involvement in depression phenotypes. Studies generally show that demethylation of *Fkbp5*, especially observed in patients with history of childhood adversity [46], and the resulting higher *Fkbp5* levels in blood contribute to increased stress sensitivity and altered emotional processing [47].

In cluster 3, with normal DST results, we observe upregulation of both *Crh* and *Crhbp* expression; however, the basal level of corticosterone remains unchanged. It can be supposed that *Crhbp* buffers excessive *Crh* activity, limiting overstimulation of *Crh* receptors, thereby preventing the development of stress-related phenotypes. However, at this point, we have little molecular data to explain the observed behavioral phenotype. In any case, balanced regulation of the HPA axis corresponds with the absence of pronounced behavioral abnormalities, suggesting that this cluster represents an adaptive response to chronic stress.

Basal expression of HPA-axis-related genes showed only subtle changes, suggesting that differences in stress reactivity may be mediated by epigenetic or other regulatory mechanisms of gene expression, rather than changes in basal gene expression. For example, DNA methylation has been implicated in chronic stress, although promoter-specific data is limited. Another possibility is that potential differences in gene expression may emerge under challenge rather than at baseline, and since all the tissues were collected after chronic stress exposure without additional acute stress or dexamethasone administration, we only detect slight alterations.

Both animal and human studies highlight the critical role of the HPA axis in stress response and psychopathology. Our findings suggest that the degree of HPA axis dysregulation may correlate with the severity of behavioral and physiological symptoms. The complex interplay between stress responses and glucocorticoid sensitivity/resistance represents an important area in understanding the pathophysiology of stress-related disorders, including depression [48].

It is important to note that depression remains a uniquely human disorder. Not all symptoms observed in patients with MDD can be replicated in animal models. However, stress-induced animal models of depression, such as CSDS, serve as valuable tools for research, enabling the investigation of molecular mechanisms in the brain and the analysis of response heterogeneity in large cohorts.

Another limitation is that synthetic glucocorticoids used in the DST differ from endogenous ones in terms of pharmacokinetics, bioavailability, blood–brain barrier penetration, and mineralocorticoid receptor cross-reactivity [49,50]. Consequently, when studying stress responses using synthetic glucocorticoids, tissue reactions may differ from those occurring under natural stress conditions.

This work highlights the importance of considering heterogeneity in stress responses. It underscores that depression cannot be accurately represented by a single physiological model, as multiple patterns of stress response exist. Future research may focus on deeper analysis of individual differences in chronic stress responses.

## 4. Materials and Methods

### 4.1. Animals

Adult male mice of the C57BL/6 and CD1 strains were provided by the Center for Genetic Resources of Laboratory Animals at the Institute of Cytology and Genetics, SB RAS, Novosibirsk, Russia (RFMEFI62119X0023). The animals were housed under standard conditions (12:12 h light/dark cycle, lights on at 7.00 A.M.; feed—pellets—and water were available ad libitum). The mice were weaned at 1 month of age and housed in groups of 8–10 in plastic cages (36 × 23 × 12 cm). Experiments were performed on mice 12 weeks of age.

### 4.2. Chronic Social Defeat Stress

A prolonged social defeat experience was induced in male mice using a sensory contact model [51] with certain modifications [7]. The mice were kept in a metal cage (14 × 28 × 10 cm) divided in half by a perforated transparent partition, allowing the animals to see, hear, and smell each other, but preventing physical contact. Male C57BL/6 mice were placed in an empty compartment next to a larger male CD1 mouse on the other side of the partition, and the pairs were left to adapt for two days. Each day, in the afternoon (from 2:00 to 5:00 p.m. local time), the partition was removed for five minutes to allow aggressive interactions. Aggressive encounters between males were halted by lowering the partition if strong aggression continued for more than one minute. During each confrontation session, the C57BL/6 mouse was attacked by the aggressive CD1 mouse and exhibited defensive behaviors (such as sideways postures, upright stances, retreat, or freezing). Once a day, after the defeat experience, each C57BL/6 mouse was placed in a new cage with a different aggressive CD1 male behind the partition. Each CD1 male remained in its home cage.

As a control C57BL/6, male mice were housed in experimental cages with the partition for 7–10 days without a partner. This control setup is most appropriate for this model [52].

### 4.3. Experimental Design

We conducted two experiments using different sets of behavioral tests.

#### 4.3.1. Experiment 1

In this experiment (Figure 1A, left), a group of male mice (33 males in total) was subjected to 31 days of chronic social defeat stress. Two control groups were included: 15 animals that underwent behavioral testing, and an intact control group (10 animals) without any interventions. To evaluate the status of the HPA axis, the dexamethasone test was performed at two time points—after 10 and 30 days of chronic stress. Behavioral tests, including the elevated plus maze, open field, social interaction, and partition tests, were conducted on days 25–28 of the stress protocol. On day 24 daily social defeat procedure was videotaped for the analysis of victim behavior.

#### 4.3.2. Experiment 2

To confirm the reliability and reproducibility of the results from Experiment 1, a second experiment was conducted with a similar design and additional behavioral assessments (Figure 1, left). A group of male mice (40 males) underwent 34 days of chronic social defeat stress. The experiment also included two control groups—with and without behavioral testing. The dexamethasone test was conducted around the 30th day of stress exposure. Additional behavioral tests included the sucrose preference test (conducted on days 28–30) and the forced swim test (on day 34). One day after the last CSDS procedure and forced swim test animals were sacrificed by decapitation, adrenal glands, hypothalamus and pituitary gland were collected for RNA extraction and subsequent qPCR analysis. All biological samples were rapidly frozen in liquid nitrogen, and stored at −70 °C until use.

### 4.4. Dexamethasone Suppression Test (DST)

The dexamethasone test involved two stages of blood sampling. To measure basal corticosterone levels, animals received an intraperitoneal injection of saline at 9:00 a.m. local time (Zeitgeber time 2), and 6 h after that (at 15:00 p.m., Zeitgeber time 8), blood samples were collected (around 100 µL). The second blood collection was performed 48 h later in Experiment 1 and 5 days later in Experiment 2. At 9:00 a.m. local time, animals were administered dexamethasone (0.1 mg/kg) 6 h prior to blood sampling. A concentration of 0.1 mg/kg dexamethasone was used to achieve partial suppression (approximately 50%) of the HPA axis [13]. In Experiment 1, blood was collected from the facial vein, while in Experiment 2, sampling was performed from the retro-orbital sinus to improve success rates, achieving up to 90%. Each sampling procedure took between 30 and 120 s per animal.

On the days when the dexamethasone test was performed (15:00–16:00), CSDS sessions were conducted only after blood sampling (at 17:00), ensuring that the chronic stress protocol itself did not acutely influence DST outcomes. We examined whether the duration of collection correlated with corticosterone concentrations, and no significant association was found.

### 4.5. Behavioral Tests

#### 4.5.1. Elevated Plus Maze

Elevated plus-maze test was used to assess anxiety-like behavior in animals [53]. The maze consisted of two opposite open arms (25 cm × 5 cm) and two opposite closed arms (25 cm × 5 cm × 15 cm), raised 50 cm above the floor. All measurements were conducted in a dimly lit experimental room. Each mouse was observed for 5 min, with the time spent in open arms, closed arms, and the central platform recorded as a percentage of the total test time, additionally the number of entries in arms of the maze was counted. The apparatus was thoroughly cleaned and dried after each animal.

#### 4.5.2. Open Field

The Open Field test was used to assess locomotor activity and individual exploratory behavior in mice. The open field arena was a brightly lit square (80 cm × 80 cm) with a white floor and 25 cm high walls. The mouse was placed at the center, and behavioral parameters were recorded for 5 min, including total explored area, time spent in the central zone (40 cm × 40 cm), and number of rearing events. The arena was thoroughly cleaned and dried after each test.

#### 4.5.3. Social Interaction Test

The Social Interaction test was conducted in a neutral arena to assess social behavior, including social investigation and approach/avoidance responses [54]. For this test, a square plastic arena (40 cm × 40 cm × 25 cm) was used. A small perforated plastic box (10 cm × 10 cm), the target zone, was placed along one wall, equidistant from the corners. The test consisted of two 5 min trials. At the start of each trial, the test mouse was placed opposite the target zone, facing the wall. During the first 5 min, the plastic box was empty, allowing the mouse to acclimate. In the second 5 min, an unfamiliar aggressive CD1 male mouse was placed in the target zone. The test mouse’s social responses were observed, including time spent in the target zone (1 cm around the box), latency to first contact, and rearing events.

#### 4.5.4. Partition Test

The Partition test was used to assess social behavior in the home cage under familiar conditions [29]. The test was conducted in a metal home cage (14 cm × 28 cm × 10 cm) divided by a transparent partition. In the first 5 min trial, a familiar aggressive CD1 male, kept in the same cage as the test mouse for the past 24 h, was behind the partition. In the second 5 min trial, an unfamiliar nonaggressive intact CD1 male was introduced behind the partition. Observations included latency to first approach, time spent near the partition, rearing, and grooming.

#### 4.5.5. Social Defeat Session

In order to assess defensive behavior strategies during aggressive interactions, behavior of defeated mice were recorded in a home cage as part of everyday routine. On day 24 the partition was removed for five minutes to allow aggressive interactions and the procedure was videotaped. We assessed the duration of freezing in stressed mice in different situations during the session: during direct attacks by the CD1 aggressor, during non-aggressive contact (sniffing), being near the aggressor (2 cm between mice) and away from the aggressor. The proportion of freezing time in each situation was calculated as a percentage of the total freezing time for each mouse.

#### 4.5.6. The Forced Swimming Test

The Forced Swim Test was used to evaluate behavioral despair, a measure commonly interpreted as an indicator of depressive-like behavior [55]. Each mouse was placed in a warm water-filled (37–38 °C, three-quarters full) transparent cylinder (height 45 cm, diameter 10 cm) and observed for 5 min. Post-test, mice were dried with paper towels and returned to their home cages. Behaviors recorded included active swimming, total immobility time, and latency to first immobility.

#### 4.5.7. The Sucrose Preference Test

The Sucrose Preference Test, based on a free-choice two-bottle paradigm, was used to assess anhedonia, which is a symptom of depressive-like behavior in rodents [56]. For three days, each mouse was given a choice between two identical bottles: one containing clean water and the other a 1% sucrose solution. The bottles were weighed once daily, and then mean values across 3 days were calculated for total liquid consumption and sucrose preference percentage (100/total_liquid_consumption × sucrose_consumption).

### 4.6. Enzyme-Linked Immunosorbent Assay

To measure corticosterone concentrations, blood samples were centrifuged twice for 10 min at 3000 g and the resulting serum was stored at −70 °C. 20 µL of serum was used for analysis. ELISA was conducted according to the manufacturer’s protocol. Experiment 1 used the Corticosterone (Human, Rat, Mouse) Kit (RE52211, IBL, Hamburg, Germany), and Experiment 2 used the ELISA Kit for Corticosterone (CEA540Ge, CLOUD-CLONE CORP., Wuhan, China). Fluorescence levels were measured using a VICTOR3 spectrophotometer (Perkin Elmer, Shelton, CT, USA).

### 4.7. RNA Extraction

RNA was extracted from frozen tissue, collected in Experiment 2, using PureZOL reagent (Bio-Rad, Hercules, CA, USA) following the manufacturer’s instructions. Total RNA was treated with RNase-free DNase I (Thermo Fisher Scientific, Waltham, MA, USA) to remove genomic DNA contamination. The isolated total RNA was purified using Agencourt RNAClean XP magnetic beads (Beckman Coulter, Krefeld, Germany). RNA quality and quantity were assessed with a NanoDrop 2000 spectrophotometer (Thermo Fisher Scientific, USA).

### 4.8. qPCR Analysis

For reverse transcription, the RevertAid First Strand Synthesis Kit (Thermo Fisher Scientific, Waltham, MA, USA) was used with 1 µg of RNA per reaction to obtain cDNA which was used for qPCR analysis of gene expression. PCR reactions were conducted in 20 µL volumes containing 0.25 mM dNTPs, 2 µL of 10× PCR buffer-B, 10 pmol of forward and reverse primers, 10 pmol of TaqMan probe, 20 ng of cDNA, and 0.3 U of SynTaq DNA polymerase with antibody-mediated enzyme inhibition (Syntol, Moscow, Russia). The protocol included initial heating at 95 °C for 5 min, followed by 39 cycles (denaturation at 95 °C for 15 s, annealing and extension at 60 °C for 20 s). Primers and probes can be found in Appendix A.

Candidate reference genes *Gpm6b* (Glycoprotein M6B) and *Hk1* (Hexokinase 1) (Appendix A) were verified using UCSC BLAT to ensure no pseudogenes or non-unique sequences were present. Stability testing of expression in the extended sample groups was performed with Bio-Rad’s CFX Manager software version 3.1, and *Gpm6b* and *Hk1* were chosen with a group coefficient of variation (CV) of 0.029 and M-value of 0.083.

### 4.9. Statistical Analysis

Behavioral events for elevated plus maze, social interaction, partition and forced swim tests were tracked using the BORIS software version 7.13.6 [57]. Open field parameters (total explored area, time spent in the central zone, and number of rearing events) were automatically assessed using EthoStudio software version 2.0 [58], with rearing and grooming events logged in BORIS software version 7.13.6.

Statistical analyses were performed separately for distinct families of outcomes, including behavioral measures (open-field, elevated plus maze, partition test, social interaction) and HPA-axis-related genes in hypothalamus, pituitary, and adrenal glands.

Each parameter was tested for normality using the Shapiro–Wilk test. Since most data followed a normal distribution, one-way ANOVA, and two-way ANOVA for tests with two trials were used. Within each variable, multiple comparisons were performed using post hoc Tukey’s HSD, which corrects for multiple testing across all pairwise comparisons within that parameter. No formal correction for multiple testing was applied across different families of outcomes or across multiple variables within a family, which is justified by the biologically guided structure of our analysis and the relatively small number of related measures.

Factors used in the study included: 1. Behavioral testing: Yes/No. 2. Experiment: 1/2. 3. Group: Control (Cntrl)/CSDS. 4. Stress duration: 0 days (control)/10 days/30 days. 5. Treatment: Dex (dexamethasone)/Sal (saline). 6. Partner type: familiar/unfamiliar. 7. Trial: 1/2.

All the factors are mentioned when described in the paper in each specific case. Statistical significance was set at *p* < 0.05. All analyses were conducted in RStudio using the stats package (v4.2.2).

### 4.10. Clustering

Animals from the CSDS group were clustered based on their dexamethasone suppression test results after 30 days of stress, using baseline corticosterone levels and levels measured 6 h post-dexamethasone injection. Only animals with complete data for both variables were included in the analysis.

Prior to clustering, corticosterone values were Z-scaled to ensure equal weighting. Clustering was performed using k-means implemented in R (cluster v.2.1.8 and factoextra v.1.0.7 packages). The number of clusters (k = 3) was chosen based on the elbow method, which evaluates the within-cluster sum of squares to identify the optimal number of clusters.

To ensure reproducibility, a fixed random seed (set.seed(5)) was used, and 25 random initial configurations were applied (nstart = 25) to reduce the likelihood of convergence to a local minimum. Cluster assignments were subsequently visualized using fviz_cluster from the factoextra package v.1.0.7.

## 5. Conclusions

Here, we identified and described clusters of mice previously subjected to CSDS that represent different coping strategies and have different HPA axis responses to chronic stress. The differentiation of stress responses and their underlying molecular features suggests that standard treatments may be less effective without considering individual stress response patterns. A personalized pharmacotherapy approach, based on the type of stress adaptation, could improve treatment efficacy and outcomes, guiding the development of new therapeutic strategies to restore normal stress hormone regulation.

## Figures and Tables

**Figure 1 ijms-26-11436-f001:**
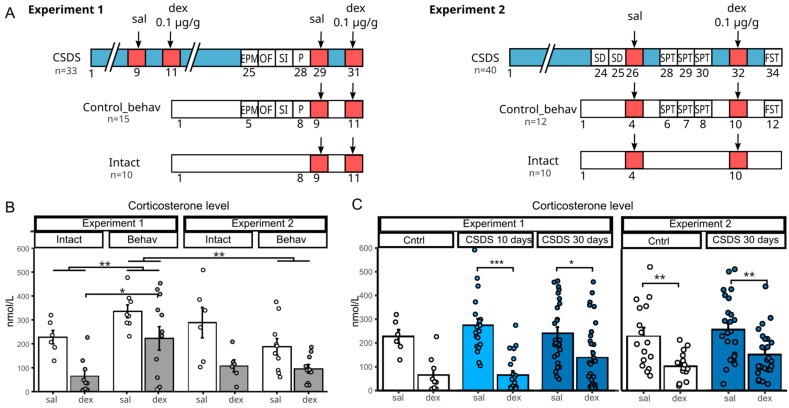
(**A**) Experimental design for Experiments 1 and 2. Arrows indicate the days on which saline or dexamethasone was administered (dexamethasone suppression test). Blood samples were collected six hours after administration on the same day. EPM—elevated plus maze, OF—open field, SI—social interaction, P—partition, FST—forced swim test; SPT—sucrose preference test; SD—social defeat test; CSDS—group of male mice was subjected to 30 days of chronic social defeat stress; Control_behav—animals without CSDS experience but with behavioral testing; Intact—animals without CSDS or behavioral testing experience. (**B**) Plasma corticosterone levels in control with behavioral testing experience (“behav”) and intact (“intact”) animals in Experiments 1 and 2. Corticosterone levels were assessed using a dexamethasone test 6 h after administration of saline (at 15:00, Zeitgeber time 8). Three-way ANOVA, TukeyHSD, * *p* adj < 0.05, ** *p* adj < 0.01. (**C**) Evaluation of HPA axis state in the dexamethasone suppression test after 10 and 30 days of chronic social defeat stress. Left panel: Experiment 1. Right panel: Experiment 2. Two-way ANOVA, TukeyHSD, * *p* adj < 0.05, ** *p* adj < 0.01, *** *p* adj < 0.001.

**Figure 2 ijms-26-11436-f002:**
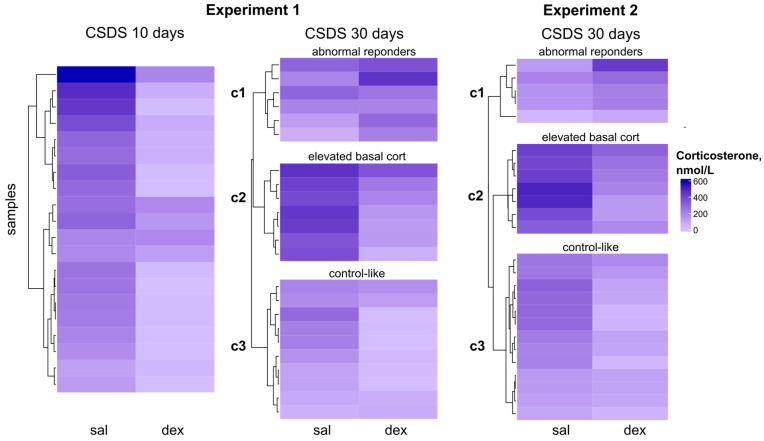
Heatmaps of plasma corticosterone concentrations (nmol/L) in C57BL/6 mice after chronic social defeat stress. From left to right: after 10 days of CSDS in Experiment 1, after 30 days of CSDS in Experiment 1, and after 30 days of CSDS in Experiment 2. Lower concentrations are shown in lighter blue; higher concentrations in darker blue. CSDS30 heatmaps separated into clusters discussed below. Abbreviations: sal—saline, dex—dexamethasone, CSDS—chronic social defeat stress. c1 (Cluster 1): low basal corticosterone + abnormal response to dexamethasone (paradoxical increase). c2 (Cluster 2): high basal corticosterone + blunted suppression after dexamethasone (≤2-fold decrease). c3 (Cluster 3): low basal corticosterone and control-like suppression after dexamethasone.

**Figure 3 ijms-26-11436-f003:**
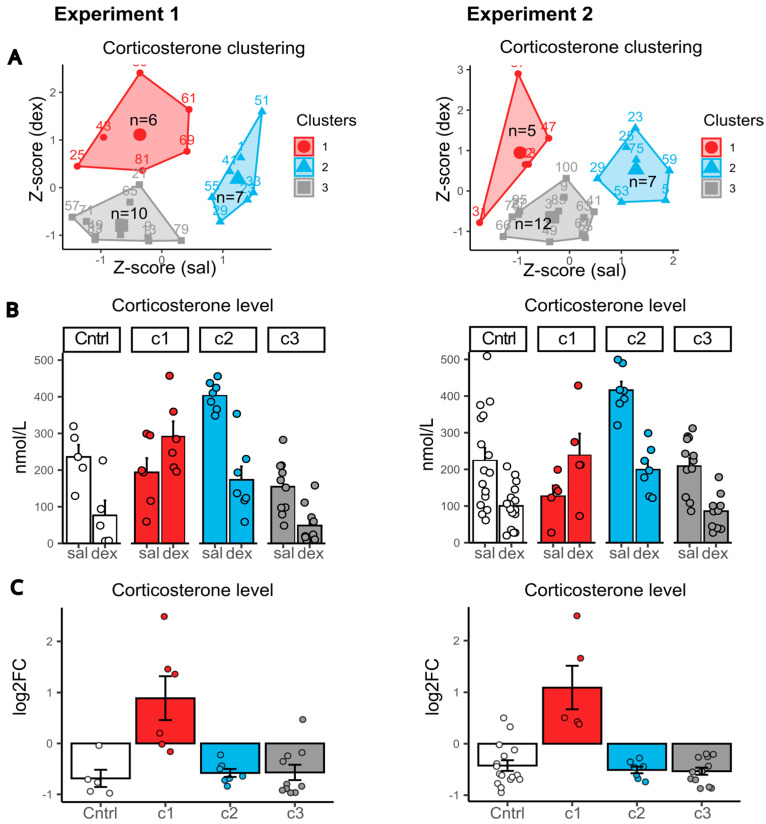
Comparative evaluation of HPA axis state in the dexamethasone suppression test after 10 and 30 days of chronic social defeat stress. Left panel: Experiment 1. Right panel: Experiment 2. (**A**) Clustering of CSDS mice by response to the DST using Z-scaled values of corticosterone serum levels after saline and dexamethasone injection. Numbers on the edges of the polygons are mouse IDs. (**B**) Corticosterone level by cluster, nmol/L. (**C**) Change in corticosterone level by cluster, log2FC—log2 fold change in corticosterone levels (dexamethasone vs. saline injection). Clusters c1, c2 and c3 are CSDS subgroups identified based on k-means cluster analysis by response to dexamethasone administration. Abbreviations: sal—saline, dex—dexamethasone, CSDS—chronic social defeat stress. c1 (Cluster 1): low basal corticosterone + abnormal response to dexamethasone (paradoxical increase). c2 (Cluster 2): high basal corticosterone + blunted suppression after dexamethasone (≤2-fold decrease). c3 (Cluster 3): low basal corticosterone and control-like suppression after dexamethasone.

**Figure 4 ijms-26-11436-f004:**
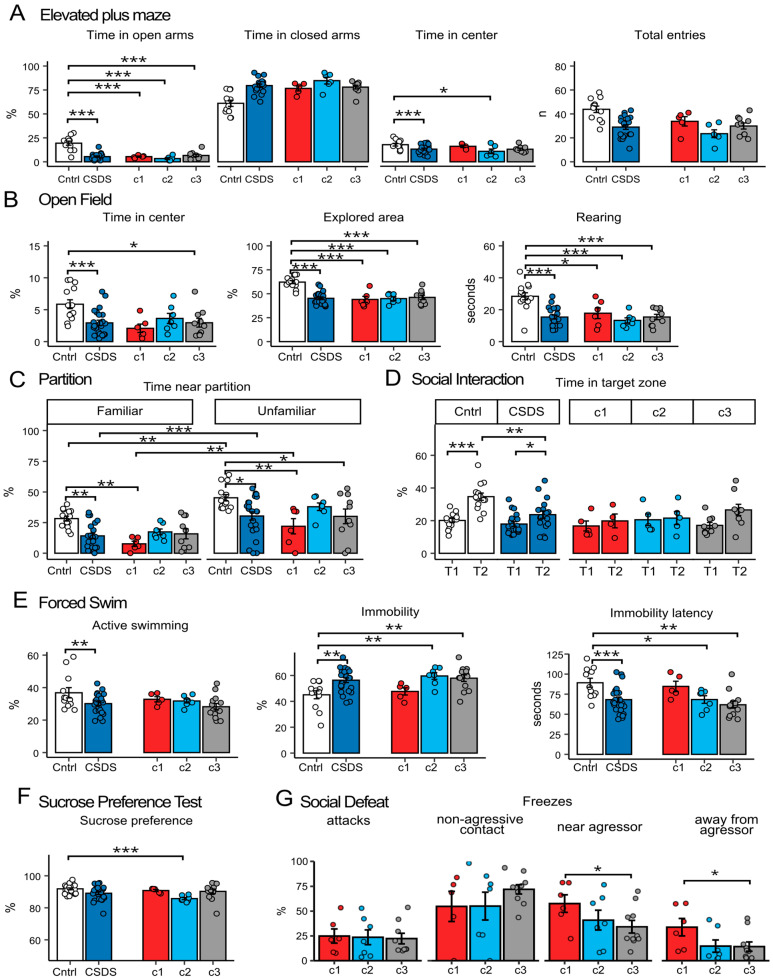
Characteristics of mice behavior after chronic social defeat stress exposure. (**A**) Elevated plus maze. (**B**) Open field. (**C**) Partition. Familiar partner—male CD1 with whom the experimental mice were kept in the home cage for the last day, unfamiliar—intact CD1 males. (**D**) Social interaction test. In the first trial, testing was carried out without a social partner, in the second trial, CD1 aggressor was placed. t1—trial 1, t2—trial 2. (**E**) Forced swimming test. (**F**) Sucrose preference test. (**G**) Social defeat session. The parameters include the percentage of total freezes occurring during the aggressor’s attack, during non-aggressive contact, time spent near or away from the aggressor. Clusters c1, c2 and c3 are CSDS subgroups identified based on k-means cluster analysis by response to dexamethasone administration. c1 (Cluster 1): low basal corticosterone + abnormal response to dexamethasone (paradoxical increase). c2 (Cluster 2): high basal corticosterone and blunted suppression after dexamethasone (≤2-fold decrease). c3 (Cluster 3): low basal corticosterone and control-like suppression after dexamethasone. One- or two-way ANOVA, TukeyHSD, * *p* adj < 0.05, ** *p* adj < 0.01, *** *p* adj < 0.001.

**Figure 5 ijms-26-11436-f005:**
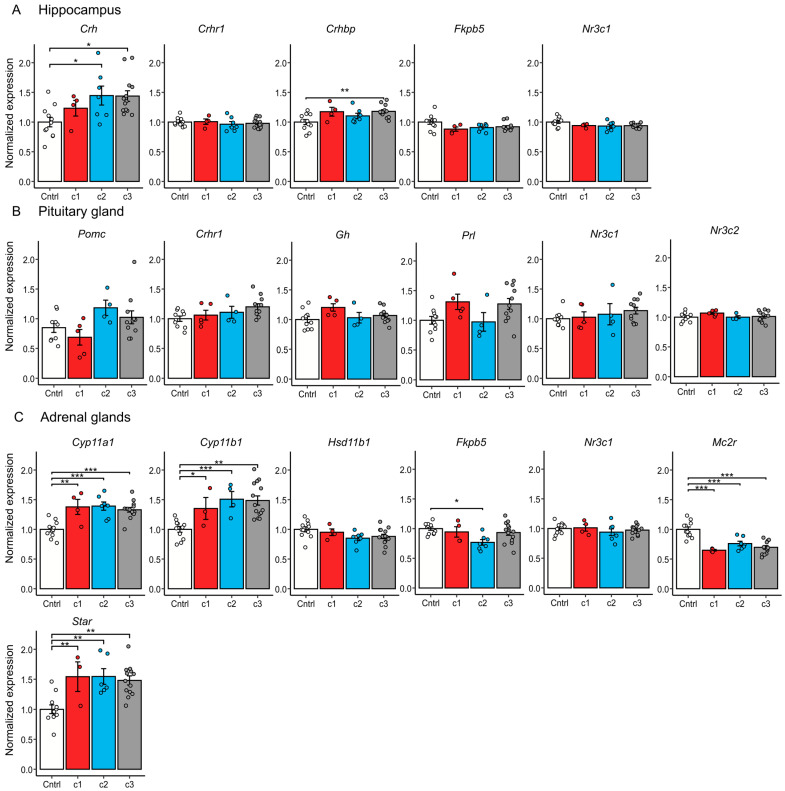
Expression of key HPA axis genes in subgroups of animals identified after CSDS exposure as shown by qPCR analysis. (**A**) Hypothalamus. (**B**) Pituitary gland. (**C**) Adrenal glands. Clusters c1, c2 and c3 are CSDS subgroups identified based on k-means cluster analysis by response to dexamethasone administration. c1 (Cluster 1): low basal corticosterone + abnormal response to dexamethasone (paradoxical increase). c2 (Cluster 2): high basal corticosterone and blunted suppression after dexamethasone (≤2-fold decrease). c3 (Cluster 3): low basal corticosterone and control-like suppression after dexamethasone. One-way ANOVA, TukeyHSD, t—*p* adj < 0.1, * *p* adj < 0.05, ** *p* adj < 0.01, *** *p* adj < 0.001.

## Data Availability

The original contributions presented in this study are included in the article/Appendix A. Further inquiries can be directed to the corresponding authors.

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
