# Peer review of "Chronic Stress Segregates Mice into Distinct Behavioral Phenotypes Based on Glucocorticoid Sensitivity"

_ijms, 2025, doi:10.3390/ijms262311436_

Round 1

Reviewer 1 Report

Comments and Suggestions for Authors

This study aims to explore the effects of chronic social defeat stress on the HPA axis by assessing corticosterone levels, behavioral phenotypes, and conducting qPCR analysis. While the key findings are significant and merit publication, the manuscript requires substantial revision before it can be considered for acceptance.

  1. Notably, Figure 4 is not referenced anywhere in the main text, and it is unclear whether the description in section 2.3 pertains solely to Figure 3. Such oversights are unexpected from an experienced research team.
  2. Several figures lack professional presentation. For instance, the y-axis label in Figure 4B only indicates the unit (nmol/L) without specifying the measured parameter—is it corticosterone? A clearer labeling convention would be, for example, “Corticosterone level (nmol/L)” or “Time (seconds).” This issue recurs in multiple figures.
  3. Consistency in labeling is essential. Each panel’s labels—including panel titles, x-axis, and y-axis—should follow the same style, such as capitalizing the first letter uniformly.
  4. The font size of tick labels in Figures 5 and 6 should be increased for better readability.
  5. Given the journal’s preference for presenting Results before Methods, the authors should consider repositioning Figure 7 ahead of Figure 1 or combining them into a new Figure 1. This adjustment would help readers better understand the study design and behavioral outcomes from the outset.
  6. Adding subheadings within section 3 would improve the manuscript’s organization and clarity.
  7. The manuscript requires thorough language refinement. For example, the title could be more concise and impactful, such as: “Chronic Stress Segregates Mice into Distinct Behavioral Phenotypes Based on Glucocorticoid Sensitivity.” Additionally, minor grammatical corrections are needed—for instance, inserting “the” before “variability” on line 15 and adding a comma between “axis” and “dexamethasone” on line 17. Overall, the abstract should be reorganized to present the background, methods, key findings, and potential implications more logically and succinctly.

Author Response

We sincerely thank the reviewer for their careful evaluation and constructive comments.

Comments 1: Notably, Figure 4 is not referenced anywhere in the main text, and it is unclear whether the description in section 2.3 pertains solely to Figure 3. Such oversights are unexpected from an experienced research team.
Response 1: Thank you for pointing this out. We agree that the lack of an explicit reference to Figure 4 in the main text was an oversight. We have now corrected this and added the appropriate citation to Figure 4 in Section 2.3 (as well as any other relevant parts of the text). We appreciate the reviewer’s careful attention to detail.

Comments 2: Several figures lack professional presentation. For instance, the y-axis label in Figure 4B only indicates the unit (nmol/L) without specifying the measured parameter—is it corticosterone? A clearer labeling convention would be, for example, “Corticosterone level (nmol/L)” or “Time (seconds).” This issue recurs in multiple figures.
Response 2: We appreciate the reviewer’s comment regarding figure presentation. We have carefully revised all relevant figures to ensure clear and professional labeling. In particular, we moved “Corticosterone level” from the y-axis into the panel titles and retained only the units (e.g., nmol/L) on the axes in Figure 1 and 2 (now combined in Figure 1). And added titles to each plot in Figure 4 (now Figure 3). This formatting has now been applied consistently across all figures.

Comments 3: Consistency in labeling is essential. Each panel’s labels—including panel titles, x-axis, and y-axis—should follow the same style, such as capitalizing the first letter uniformly.
Response 3: We thank the reviewer for this observation. We have carefully checked all figure panels and ensured that axis labels, panel titles, and overall formatting follow a consistent style. In particular, we moved “Сorticosterone level” from the y-axis into the panel titles and kept only the units (e.g., nmol/L) on the axes. Units (e.g., nmol/L, s, log2FC) are presented in lowercase in accordance with standard scientific notation. We now believe the figure formatting is fully consistent throughout the manuscript.

Comments 4 and 5: The font size of tick labels in Figures 5 and 6 should be increased for better readability.
Given the journal’s preference for presenting Results before Methods, the authors should consider repositioning Figure 7 ahead of Figure 1 or combining them into a new Figure 1. This adjustment would help readers better understand the study design and behavioral outcomes from the outset.
Response 4 and 5: We fully agree with the reviewer’s suggestion. In response, we have reorganized the figures by merging the experimental design with the original Figures 1 and 2. As a result, the study design and the key corticosterone findings are now presented together in a new Figure 1, allowing readers to grasp the experimental context and main outcomes from the outset.
The tick label font size in Figures 5 and 6 (now Figures 4 and 5, respectively) has been increased to improve readability. To reduce figure density, we moved several plots from Figure 4 (behavioral tests) to Supplementary Figure S1, including latency to first approach, rearing, and grooming time from the partition test; latency to first approach and rearing from the social interaction test; and total liquid consumption from the sucrose preference test.

Comments 6: Adding subheadings within section 3 would improve the manuscript’s organization and clarity.
Response 6: We thank the reviewer for this suggestion. We have now added informative subheadings within Section 3 to improve the organization and clarity of the manuscript. The subheadings are as follows:
3.1.Dynamics of HPA axis responsiveness during chronic social stress
3.2.Heterogeneity of stress responses and their possible correspondence to MDD subtypes
3.3.Transcriptional signatures associated with differential HPA axis responses
These subheadings help guide the reader through the section, highlighting the temporal dynamics, behavioral clusters, and molecular analyses.

Comments 7: The manuscript requires thorough language refinement. For example, the title could be more concise and impactful, such as: “Chronic Stress Segregates Mice into Distinct Behavioral Phenotypes Based on Glucocorticoid Sensitivity.” Additionally, minor grammatical corrections are needed - for instance, inserting “the” before “variability” on line 15 and adding a comma between “axis” and “dexamethasone” on line 17. Overall, the abstract should be reorganized to present the background, methods, key findings, and potential implications more logically and succinctly.
Response 7: We appreciate the reviewer’s thoughtful comments. We have adopted the suggested and much improved article title. We also agree that the abstract required reorganization, and we have revised it to present the background, methods, main findings, and implications more clearly and coherently. In addition, we have revised the manuscript for clarity and readability and corrected the grammatical issues.

Reviewer 2 Report

Comments and Suggestions for Authors

The manuscript by Ritter et al. entitled “Chronic stress leads to differentiation of mice by glucocorticoid sensitivity into groups characterized by specific behavioral phenotype” addresses an interesting question about heterogeneity in stress responses and HPA axis regulation. However, the manuscript in its current form has drawbacks that must be resolved before it can be considered for publication.

Major points:

  1. The attempt to map the three CST/DEX-response clusters onto human MDD subtypes (melancholic/atypical, mild/moderate depression) is interesting but sounds speculative. The behavioral phenotype in mice only partially reflects human symptom clusters, and the neuroendocrine readouts are limited to corticosterone and selected HPA-related genes. The authors should discuss these translational claims more carefully and provide a more systematic justification, for example provide information aligning cluster features with canonical DSM/clinical features and neuroendocrine patterns, while clearly stating limitations.
  2. The clustering of animals based on DST corticosterone levels is central to the study but not described in sufficient technical detail or validated robustly. The authors should provide a full description of the clustering procedure: exact variables used, scaling method, software applied, random seeding, number of iterations, and how missing values were handled.
  3. The rationale for the dex dose (0.1 mg/kg) is given, but the observation of paradoxical increases in some animals raises questions about pharmacokinetics and timing. Please discuss whether stress from blood sampling, handling, and the chronic CSDS protocol itself could influence the DST outcome.
  4. The authors should clarify the timing of DST relative to the last defeat session and behavioral testing. Could acute stress or behavioral test exposure immediately preceding the DST contribute to cluster segregation?
  5. Some conclusions (cluster 1 “resembling atypical depression”, cluster 2 “melancholic-like”) should be better grounded in the actual behavioral data or rephrased as hypotheses. For example, cluster 1 animals show low social activity and freezing but do not appear clearly atypical in a human sense.
  6. Due to the fact that the cluster sizes are modest, please discuss stability: whether similar clusters arise when using experiment 1 and experiment 2 separately, or via bootstrapped clustering.
  7. Regarding statistical analysis, the study reports numerous ANOVAs across many behavioral parameters and multiple genes. The authors should clarify whether any correction for multiple testing was applied across families of related outcomes (for example, all EPM variables, all open-field variables, all hypothalamic genes, all adrenal genes). If not, the authors should justify their decision and consider applying some type of correction.
  8. When reporting “trends” (for example, p-values 0.07-0.1), authors should be cautious. Either clearly define in Methods what you consider a “trend” or avoid this terminology and focus on effect sizes and confidence intervals.
  9. Some of the mechanistic interpretations (for example, Crh/Crhbp balance in cluster 3 as an adaptive mechanism, Fkbp5 changes in cluster 2 leading to sustained hypercortisolemia) should be made more cautious and explicitly labelled as hypotheses, unless supported by additional functional data.
  10. Some figures (especially heatmaps and multi-panel behavioral plots) are dense. Consider simplifying figure panels or adding more detailed legends and/or inset schematics to make them more readable.

Author Response

We greatly appreciate the reviewer’s insightful feedback and thoughtful suggestions.

Comments 1:The attempt to map the three CST/DEX-response clusters onto human MDD subtypes (melancholic/atypical, mild/moderate depression) is interesting but sounds speculative. The behavioral phenotype in mice only partially reflects human symptom clusters, and the neuroendocrine readouts are limited to corticosterone and selected HPA-related genes. The authors should discuss these translational claims more carefully and provide a more systematic justification, for example provide information aligning cluster features with canonical DSM/clinical features and neuroendocrine patterns, while clearly stating limitations.
Response 1: Thank you very much for this thoughtful and important comment. We fully agree that mapping the CST/DEX-response clusters onto human MDD subtypes is inherently speculative and should be interpreted with caution. Our intention was simply to highlight conceptual parallels in HPA-axis heterogeneity, not to imply a direct translational equivalence.
In the revised manuscript, we now explicitly acknowledge these limitations and clarify that this comparison is hypothetical and intended only as a heuristic framework. Together, these additions ensure that the discussion remains grounded and appropriately cautious while preserving the value of considering heterogeneity in stress responses. 
In addition, following the recommendation of Reviewer 1, we have improved the manuscript’s English and adopted the new article title: “Chronic Stress Segregates Mice into Distinct Behavioral Phenotypes Based on Glucocorticoid Sensitivity.”
Comments 2: The clustering of animals based on DST corticosterone levels is central to the study but not described in sufficient technical detail or validated robustly. The authors should provide a full description of the clustering procedure: exact variables used, scaling method, software applied, random seeding, number of iterations, and how missing values were handled.

Response 2: We thank the reviewer for this valuable suggestion. We have now provided a full description of the clustering procedure in the Methods section, specifying the variables used, handling of missing data, Z-scaling, k-means parameters (including random seed and nstart), and that the number of clusters was chosen based on the elbow method, which evaluates the within-cluster sum of squares to identify the optimal number of clusters.

Comments 3: The rationale for the dex dose (0.1 mg/kg) is given, but the observation of paradoxical increases in some animals raises questions about pharmacokinetics and timing. Please discuss whether stress from blood sampling, handling, and the chronic CSDS protocol itself could influence the DST outcome.
Response 3: We thank the reviewer for this important point. We fully agree that acute stress can influence DST results, and we took care to minimize such effects. On DST days, blood sampling (between 15:00 and 16:00) was performed before the CSDS session, which began at 17:00. Each blood collection was brief (30–120 seconds per animal) and conducted under carefully controlled conditions. We were mindful that the sampling procedure itself could affect corticosterone levels, so we examined whether the duration of blood collection correlated with corticosterone concentrations, and no significant association was found. While acute stress cannot be entirely excluded, we carefully designed the experiment to reduce its impact and to ensure the reliability of our measurements.
One possible explanation for the pattern observed in cluster 1 is that these animals may show glucocorticoid escape, a phenomenon seen in some patients with depression where corticosterone initially responds to dexamethasone but returns to baseline earlier than expected, or can be elevated.  However, as we do not have detailed time-series data on our samples, this interpretation remains tentative. These clarifications have been added to the Methods section of the revised manuscript.

Comments 4: The authors should clarify the timing of DST relative to the last defeat session and behavioral testing. Could acute stress or behavioral test exposure immediately preceding the DST contribute to cluster segregation?
Response 4: We thank the reviewer for this very insightful observation. Acute stress immediately preceding the dexamethasone suppression test (DST) could potentially influence corticosterone measurements and cluster segregation. To minimize such effects, on the days of the DST, CSDS sessions were conducted only after blood sampling (24 hours after the previous CSDS session), ensuring that acute stress from social confrontations did not impact the results. Furthermore, the experiment was carefully designed so that the day prior to the DST involved only minimally invasive procedures: the Partition test (experiment 1) conducted in the home cage and the social defeat test (experiment 2) as part of the daily routine schedule, which are unlikely to induce additional HPA-axis activation. Details of these procedures and their timing have been added to the Methods section of the revised manuscript.

Comments 5: Some conclusions (cluster 1 “resembling atypical depression”, cluster 2 “melancholic-like”) should be better grounded in the actual behavioral data or rephrased as hypotheses. For example, cluster 1 animals show low social activity and freezing but do not appear clearly atypical in a human sense.
Response 5: We thank the reviewer for this insightful comment. In response, we have removed the definitive statements from the Conclusions regarding any analogy with MDD subtypes. We added a paragraph in the Discussion emphasizing the caution required when interpreting these clusters. The descriptions have been rephrased more cautiously to highlight that any analogy with human depression subtypes is conceptual and is not to be interpreted as direct correspondence.

Comments 6: Due to the fact that the cluster sizes are modest, please discuss stability: whether similar clusters arise when using experiment 1 and experiment 2 separately, or via bootstrapped clustering.
Response 6: We thank the reviewer for this important suggestion regarding cluster stability. In the present study, clustering was performed independently in Experiment 1 and Experiment 2. Despite these being fully separate cohorts with slightly different behavioral batteries, both datasets yielded a similar three-cluster solution based on basal and dexamethasone-suppressed corticosterone levels. This replication across independent experiments already provides strong support for the robustness of the clusters.

To further explore cluster stability, we performed bootstrap resampling analyses (k-means, 100 iterations) on both the combined dataset and the independent experiments. The results showed the following mean Jaccard indices for each cluster:
Combined data: Cluster 1 (abnormal corticosterone elevation) 0.56, Cluster 2 (elevated basal corticosterone) 0.81, Cluster 3 (normal response) 0.80
Experiment 1: Cluster 1 0.60, Cluster 2 0.76, Cluster 3 0.82
Experiment 2: Cluster 1 0.68, Cluster 2 0.82, Cluster 3 0.85
Clusters 2 and 3 were highly stable, whereas cluster 1 appeared less stable in some iterations. 
The lower stability of cluster 1 (abnormal corticosterone elevation) in the bootstrap analyses reflects its small sample size and the rarity of this phenotype, rather than a flaw in the clustering procedure. In bootstrap resampling, each iteration samples with replacement from the original dataset, so small clusters may occasionally be underrepresented or entirely absent in some iterations. Consequently, the Jaccard index for this cluster is lower, even though the cluster is biologically meaningful and consistently identifiable in the original datasets.
This lower apparent stability for cluster 1 reflects its small sample size and the rarity of this phenotype, rather than a flaw in clustering. Importantly, it does not indicate that the cluster is biologically incorrect.
Given this, while we find the bootstrap analysis interesting, with your permission we have opted to focus the manuscript discussion on the reproducibility of clusters across independent experimental cohorts, which we believe provides the most biologically meaningful demonstration of cluster robustness.

Comments 7: Regarding statistical analysis, the study reports numerous ANOVAs across many behavioral parameters and multiple genes. The authors should clarify whether any correction for multiple testing was applied across families of related outcomes (for example, all EPM variables, all open-field variables, all hypothalamic genes, all adrenal genes). If not, the authors should justify their decision and consider applying some type of correction.
Response 7: We thank the reviewer for this important observation. Statistical analyses were performed separately for distinct families of outcomes, including behavioral measures (open-field, elevated plus maze, partition test, social interaction) and HPA axis-related genes in hypothalamus, pituitary, and adrenal glands. Within each family, the number of variables is relatively small (e.g., 3–4 behavioral parameters per test, 5–7 genes per structure), and main findings were interpreted in the context of these biologically defined outcome sets.
Within each variable, multiple comparisons were performed using post hoc Tukey’s HSD, which corrects for multiple testing across all pairwise comparisons within that parameter. These comparisons represented preplanned, biologically meaningful contrasts (e.g., control vs. stress, control vs. stress subclusters 1–3), rather than exploratory testing across unrelated measures. No formal correction for multiple testing was applied across different families of outcomes or across multiple variables within a family, which is justified by the biologically guided structure of our analysis and the relatively small number of related measures.

Comments 8: When reporting “trends” (for example, p-values 0.07-0.1), authors should be cautious. Either clearly define in Methods what you consider a “trend” or avoid this terminology and focus on effect sizes and confidence intervals.
Response 8: We thank the reviewer for this important point. In response, we have removed references to “trends” (p-values 0.07–0.1) from the manuscript to avoid potential misinterpretation.

Comments 9: Some of the mechanistic interpretations (for example, Crh/Crhbp balance in cluster 3 as an adaptive mechanism, Fkbp5 changes in cluster 2 leading to sustained hypercortisolemia) should be made more cautious and explicitly labelled as hypotheses, unless supported by additional functional data.
Response 9: We thank the reviewer for this suggestion. We have removed some mechanistic statements and rephrased others to be more cautious, now explicitly presenting them as hypotheses rather than established conclusions.

Comments 10: Some figures (especially heatmaps and multi-panel behavioral plots) are dense. Consider simplifying figure panels or adding more detailed legends and/or inset schematics to make them more readable.
Response 10: We thank the reviewer for this suggestion. To improve readability, we have simplified the behavioral figures by moving some panels to the Supplementary Figures. The remaining panels are now clearly labeled with the names of the tests, and spacing between plots has been increased. In addition, we relocated the experimental schematic to Figure 1 and combined it with Figure 2 to provide a clearer overview of the study design along with the results of the dexamethasone suppression test.

Round 2

Reviewer 1 Report

Comments and Suggestions for Authors

I have no other comments.

Reviewer 2 Report

Comments and Suggestions for Authors

The Authors are highly appreciated for providing accurate and comprehensive responses. The manuscript has been significantly improved and can be approved for publication in present form.